# Gastrointestinal Stromal Tumors (GISTs): Novel Therapeutic Strategies with Immunotherapy and Small Molecules

**DOI:** 10.3390/ijms22020493

**Published:** 2021-01-06

**Authors:** Christos Vallilas, Panagiotis Sarantis, Anastasios Kyriazoglou, Evangelos Koustas, Stamatios Theocharis, Athanasios G. Papavassiliou, Michalis V. Karamouzis

**Affiliations:** 1Molecular Oncology Unit, Department of Biological Chemistry, Medical School, National and Kapodistrian University of Athens, 11527 Athens, Greece; chris-vallilas@hotmail.com (C.V.); panayotissarantis@gmail.com (P.S.); vang.koustas@gmail.com (E.K.); papavas@med.uoa.gr (A.G.P.); 22nd Propaedeutic Department of Medicine, ATTIKON University Hospital, 12462 Athens, Greece; tassoskyr@gmail.com; 3First Department of Pathology, Medical School, National and Kapodistrian University of Athens, 11527 Athens, Greece; stamtheo@med.uoa.gr

**Keywords:** GIST, immunotherapy, small molecules, imatinib

## Abstract

Gastrointestinal stromal tumors (GISTs) are the most common types of malignant mesenchymal tumors in the gastrointestinal tract, with an estimated incidence of 1.5/100.000 per year and 1–2% of gastrointestinal neoplasms. About 75–80% of patients have mutations in the KIT gene in exons 9, 11, 13, 14, 17, and 5–10% of patients have mutations in the platelet-derived growth factor receptor a (PDGFRA) gene in exons 12, 14, 18. Moreover, 10–15% of patients have no mutations and are classified as wild type GIST. The treatment for metastatic or unresectable GISTs includes imatinib, sunitinib, and regorafenib. So far, GIST therapies have raised great expectations and offered patients a better quality of life, but increased pharmacological resistance to tyrosine kinase inhibitors is often observed. New treatment options have emerged, with ripretinib, avapritinib, and cabozantinib getting approvals for these tumors. Nowadays, immune checkpoint inhibitors form a new landscape in cancer therapeutics and have already shown remarkable responses in various tumors. Studies in melanoma, non-small-cell lung cancer, and renal cell carcinoma are very encouraging as these inhibitors have increased survival rates. The purpose of this review is to present alternative approaches for the treatment of the GIST patients, such as combinations of immunotherapy and novel inhibitors with traditional therapies (tyrosine kinase inhibitors).

## 1. Introduction

Gastrointestinal Stromal Tumors (GISTs) are the most common types of mesenchymal tumors of the gastrointestinal tract and originate from interstitial Cajal cells [1]. GISTs are rare tumors, with an estimated incidence of 1.5/100.000 per year and account for 1–2% of gastrointestinal neoplasms. The disease’s median age is around 60–65 years old [2,3]. The most common localization is the stomach (60%) and the small intestine (20-30%), whereas GISTs are found less frequently in ortho-sigmoid and esophagus. The main clinical manifestations are not disease-specific, including hemorrhage, anemia, indigestion, and abdominal pain due to stressful events [4,5].

About 85% of the GISTs cases are associated with a known mutation. About 75–80% of patients have mutations in the KIT gene in exons 9, 11, 13, 14, leading to a truncated c-KIT/CD117 protein, which acts as a growth factor receptor located on normal cells’ surface. Mutations in the PDGFRA gene are identified in exons 12, 14, 18 to 5–10% of patients. Approximately 10–15% of patients have no mutations and are classified as wild type GIST [6,7]. Molecular characterization of GISTs has revealed novel mutations to BRAF, neurofibromatosis type 1 (NF1), and succinate dehydrogenase (SDH) in small percentages [8]. However, even after comprehensive analysis, a group of GISTs with no mutations is identified [9].

The NIH classification system categorizes patients into very low, low, intermediate, and high-risk groups taking into consideration the size of the lesion and the mitotic activity of the tumor. The conclusion is that tumors >5 cm (diameter), plus mitotic counter higher than 5/50 high power fields (HPF) and tumors >10 cm with any mitotic rate have a high risk of recurrence and indicate adjuvant chemotherapy [10]. It was shown that the risk of recurrence is more significant for non-gastric GISTs than for gastric [11,12].

Diagnostic imaging includes computed tomography (CT), magnetic resonance imaging (MRI), and positron emission tomography (PET) has the advantage of displaying the thickness of the small bowel, leading to better visualization of deep ileal loops and mesentery [13]. Further, CT analysis facilitates the assessment of tyrosine kinase inhibitors (TKIs) response by using both RECIST and Choi criteria [14]. MRI can provide information about size, tumor perforation, or metastasis. PET scan with 2-(F-18)-fluoro-2-deoxy-d-glucose can be useful to identify areas of necrosis in tumors and differentiate benign versus malignant tumors and is used alone or with the CT for determinating the effectiveness of the adjuvant therapy [11].

Cytotoxic chemotherapy and radiation are not effective; thus, surgical resection appears as the only effective therapeutic intervention. Surgery in metastatic or recurrent is controversial, and case selection is essential. It can be helpful to patients whose neoplasm is responding to adjuvant therapy, to those with limited focal progression and as palliative surgery [15].

For metastatic or recurrent GIST, the first treatment line is the tyrosine kinase inhibitor, imatinib mesylate (IM). Patients with PDGFRA exon 18 mutation must receive avapritinib on the first line (Table 1). The therapeutic approach of GISTs for the second line of treatment contains sunitinib malate, and the third line regorafenib (with better outcomes if we choose personalized doses of regorafenib) [16,17,18].

However, the addition of ripretinib and avapritinib to our therapeutic armamentarium challenges this algorithm [19,20] (Table 2). Imatinib is a TKI that works by binding to the ATP binding sites on CD117 and PDGFRA, which leads to a block of signal transduction. Patients who have c-KIT and PDGFRA are benefited from this therapy [21].

Pediatric GISTs represent a clinically and molecularly distinct subset, characterized by the absence of c-KIT/PDGFRA mutations. Syndromes linked to GISTs are the Carney triad syndrome, Carney-Stratakis syndrome, and Neurofibromatosis type I [22].

## 2. Resistance to Imatinib

Despite the beneficial effect of tyrosine kinase inhibitors (TKIs) targeted therapy, due to the lack of complete elimination of GIST cells, eventually, 50% of the patients develop primary or secondary resistance against imatinib after two years. It should be mentioned that tumor progression is sometimes marked by an increase in density with or without an increase in the tumor size. Besides, cancer progression within the first six months after treatment with imatinib means develop of primary resistance for the patients [17,23,24].

Some GISTs lose expression of KIT oncoproteins and, therefore, become KIT-independent and, subsequently, resistant to KIT-inhibitor drugs [25]. Patients with CD117 mutations in exons 9, 11, 13, 14, 17 raise resistance to imatinib [7,26]. In 2014, patients with CD117 exon 9 mutations had better survival than patients with exon 11 mutations [27].

As primary resistance characterizes the tumor development through an initial imatinib challenge, as well as secondary resistance characterizes the tumor progresses after an initial period of response to imatinib [28]. Specific mutations and metastatic capability for specific regions have also been associated with secondary resistance (common location for GIST metastasis are liver (28%), mesentery and omentum (30%), lung (7%), subcutaneous tissues (4.7%), lymph nodes (4.7%), and bone (2.3%) [11]. Recent studies have shown that ligands from the fibroblast growth factors (FGF) family reduce imatinib’s effect on GIST cells, and FGF2 and FGFR1 are highly expressed in all primary GIST samples. Therefore, inhibitors for the FGF family constitute a potential therapy to overcome the resistance. Plenty of encouraging results exist for a number of neoplasms, e.g., bladder, digestive system, myeloma, and endometrial. These positive results give us the impetus for an extensive study of these molecules (e.g., SU5402, AZD4547, D173074, NDGA) and elucidate the pathways of GIST tumor development [29,30,31].

Epidermal growth factor receptor (EGFR) mutations identification is a rare finding in GISTs. In their publication, Shi et al. reported three EGFR mutated cases out of 323 GISTs tested, which accounted for 3.4% of the wild type GISTs of their cohort [32]. EGFR mutation was equally exclusive of any c-KIT, PDGFR, BRAF, and SDH mutations, indicating that the EGFR signaling pathway may phosphorylate and subsequently activate important downstream targets triggering proliferation and survival [32]. EGFR expression was shown from Cai et al. in GIST tumor samples, recommending an autocrine loop between transforming growth factor-α (TGF-α) and EGFR [33]. Furthermore, expression of EGFR and its ligands may be promoted by ADAMs activation, as was studied in GIST tumor samples, implying an important role of the EGFR pathway for GIST oncogenesis [34]. However, EGFR expression studies did not show any prognostic relevance to GIST patient outcomes [35]. In contrast, in their study, Zhao et al. showed that in high-risk GISTs, decreased expression of EGFR is associated with lower recurrent free survival (RFS) after treatment with imatinib [36].

The expression of EGFR has been further studied in several GIST cell lines. In a recent paper, Tu et al. have shown that in c-KIT independent GIST cell lines, EGFR was expressed, while in c-KIT positive cell lines, there was not EGFR expression [37]. Thus, it is reasonable to assume that EGFR expression is associated with imatinib resistance. However, treatment with gefitinib to c-KIT independent GIST cell lines did not impact their growth and any of the activated signals (AKT, MAPK), therefore implying that EGFR activation is not a key to imatinib resistance [37]. On the other hand, Nagata et al. showed that c-KIT and EGFR phosphorylation status is similar in imatinib-resistant GIST cell lines. In this case, the addition of gefitinib to standard imatinib treatment resulted in decreasing cell proliferation [38]. Conflicting data from in vivo experiments highlights the complexity of resistance to imatinib therapy and the unmet need to understand this process’s molecular mechanisms.

### Novel Therapies

A literature search revealed 313 clinical trials for GISTs, including more than 86 molecules studied in order to discover new effective therapies. First-line drug therapy imatinib (STI571) has been used in 152 clinical trials, sunitinib in 74, and regorafenib (BAY73-4506) in 21. Ongoing clinical trials are presented in Table 3, Table 4 and Table 5. Through clinical trials of phases I, II, III, and IV, scientists try to target GISTs with new modalities and novel molecules or combine the classic therapeutic options (imatinib) with immunotherapy (anti-PDL1 or anti-PD1).

## 3. Immunotherapy in GISTs

In recent years, there has been an increasing interest in immunotherapy-based therapeutic strategies against cancer, in attempt to incorporate all of the stepwise events required for tumor development and progression. It is well known that cancer cells, to evade the immune response, express numerous different molecules on the cell surface known, consequently leading to immune suppression. Vigorous research in the field of immunology has led to the development of a plethora of agents, also known as checkpoint inhibitors (i.e., anti-programmed death-1 (PD1)/PD-L1 and anti-cytotoxic T-lymphocyte-associated antigen 4 (CTLA4)), which are able to interrupt the inhibitory conjunction of cancer and T-cells. The checkpoint inhibitors are already used in a vast panel of cancer types [39].

The D842V mutation is the most common mutation associated with primary resistance to imatinib because it alters the kinase domain formation and, therefore, negatively affects imatinib association [40,41]. In fact, the progression free survival (PFS) of patients treated with imatinib carrying the D842V mutation compared to non-D842Vs is statistically significant (2.8 vs. 28.5 months, respectively) [42]. In another study of a cohort of patients with PDGFRA D842V-mutated GIST, imatinib resulted in progressive disease for most patients (58.9%), with PFS up to 8 months [43]. Moreover, resistance occurs due to this mutation in the sunitinib [40,41].

Many studies indicate that D824V mutation displays more immune cells with increased cytolytic activity [40,41,44,45,46,47]. In particular, express higher interferon levels and several chemokines, such as CXCL14; demonstrates additional driver-derived neoepitope-HLA binding proteins; and has more PD-1 and PD-L1 expressing tumors [40,41,47]. Differences in tumor microenvironment composition were also highlighted between the D824 mutation and the other GIST molecular subtypes. More specifically, there is an overexpression of T-regs and CD8 + T-cells and a lower CD4 + T-cell rate. Further, the T-cell-inflamed signature score for GISTs was at a similar rate with tumor types responsive to checkpoint inhibition [44,45]. All of the above data positively support the implementation of an immune-treatment approach in GIST and, in fact, in patients carrying the D824V mutation.

Seventeen clinical trials using immunotherapeutic agents in GISTs have been scheduled (Table 6). 

However, how are the new drugs combined with the older ones in the clinical trials?

Immunotherapeutic agents used in clinical trials are anti-PD-1/PDL 1 molecules and CTLA -4 (Table 7). An interim analysis of the randomized phase II clinical trial using nivolumab, a human immunoglobulin IgG4 monoclonal antibody, which is directed against the negative immunoregulatory human cell surface receptor programmed death-1 (PD-1, PDL-1) with immune checkpoint inhibitory and antineoplastic activities, with or without ipilimumab, a recombinant human immunoglobulin IgG1 monoclonal antibody, which is directed against the human T-cell receptor cytotoxic T-lymphocyte-associated antigen 4 (CTLA4), with immune checkpoint inhibitory and antineoplastic activities, in 40 patients with metastatic or inoperable GIST, was recently published [48]. In the nivolumab arm, 7 out of the 15 patients showed SD as the best response. In the combination arm, 1 out of 12 patients showed PR and 2 of the 12 patients SD. There are three more clinical trials ongoing based on the same combination, with 164 and 60 patients without any results known [49,50,51] and one phase II clinical trial that will start recruiting patients in the next few months [52]. Moreover, ipilimumab was used with dasatinib treating patients with GISTs or other sarcomas that cannot be removed by surgery or are metastatic (phase I), unfortunately with no hopeful results. Pembrolizumab (MK3475) is a humanized monoclonal immunoglobulin IgG4 antibody directed against receptor PD-1 with potential immune checkpoint inhibitory antineoplastic activities, and was used with epacadostat in one clinical trial. However, this study’s enrollment was terminated early for insufficient evidence of clinical efficacy [53]. An ongoing phase II multicenter clinical trial PEMBROSARC combines pembrolizumab with metronomic cyclophosphamide in patients with advanced sarcomas (31 patients with GIST [54]. Spartalizumab a humanized monoclonal antibody. It is directed against the negative immunoregulatory human cell surface receptor programmed death-1 (PD-1, PCD-1), which is used in one phase Ib clinical trial, not yet recruiting patients, in combination with ribociclib with available results, until now [55]. Avelumab is a humanized monoclonal anti-PD1 agent. There is an ongoing clinical trial combining avelumab with axitinib in patients with unresectable or metastatic GIST after failure of standard therapy [56]. There is another ongoing phase I/II clinical trial combining avelumab with regorafenib in patients with solid tumors and GIST [57]. Unfortunately, we have no results available yet for both clinical trials. PDR001 is a monoclonal anti-PD-1 agent. There are two clinical trials with PDR001, the first phase I/II clinical trial is combining PDR001 with imatinib in patients with advanced GIST after failure of standard TKI therapies, without any results yet [58].

## 4. Molecules-Drugs Studied in GISTs and How are They Used in Clinical Trials Now

Nilotinib, a TKI inhibitor designed to overcome imatinib resistance resulting from Bcr-Abl kinase mutations, is also a PDGF-R, c-KIT inhibitor. Nilotinib has been used in 18 clinical trials, 15 completed, two not recruiting, and one active. An ongoing phase IV clinical trial with 300 patients with GIST and chronic myeloid leukemia (CML) is trying to find if nilotinib can be useful [59]. There are two more phase IV clinical trials, active but not recruiting patients yet [60,61]. Unfortunately, the other clinical trials that were completed did not show encouraging results.

Another molecule, Ripretinib, is a selective KIT and PDGFRA inhibitor. There is a phase III clinical trial, recruiting patients with advanced GIST after treatment with imatinib. Four hundred twenty-six patients are enrolled in two arms, one with ripretinib, and the other with sunitinib; the first results will be announced after June 2021 [19]. Another ongoing phase I clinical trial with 320 patients with advanced malignancies has no results as of yet [62]. Three more clinical trials are about to begin, recruiting patients with GIST in the next few months [63,64,65].

Avapritinib is a PDGFRa and mast/stem cell factor receptor c-KIT inhibitor. There is an ongoing phase I/II clinical trial with 87 participants with unresectable or metastatic GIST with no results yet [66]. Two more clinical trials with avapritinib will start recruiting patients in the next few months [67,68]. Avapritinib was approved in 2020 by the FDA for gastrointestinal stromal tumors with a mutation in exon 18 PDGFRA (NAVIGATOR study) [69], a multicenter, single-arm, open-label trial. Forty-three patients with GIST with PDGFRA exon 18 mutation, 38 of them with PDGFRA D842V mutations. The overall response rate (ORR) was 84% (95% CI: 69%, 93%), with 7% complete responses and 77% partial responses. For patients with PDGFRA D842V mutations, the ORR was 89% (95% CI: 75%, 97%), with 8% complete responses and 82% partial responses. The median response period was not reached with a median period of record for all patients of 10.6 months (range 0.3 to 24.9 months); 61% of the responding patients with exon 18 mutations had a response lasting at least six months (31% of patients with an ongoing response were followed for less than six months).

Cabozantinib in a phase II trial has shown antitumor activity in GIST patients treated with three or more previous lines of therapy. The CaboGIST EORTC 1317 trial met its primary endpoint with 30 of the 50 patients being progression free after 12 weeks. The mPFS was 5.5 months (3.6–6.9 months) [70]. As a result of this publication, cabozantinib was included in the recent GIST NCCN guidelines as an option after failure of other approved regimens.

Moreover, Pazopanib is a vascular endothelial growth factor receptor(VEGFR)-1, -2, and -3, c-KIT inhibitor and PDGF-R inhibitor. There is no ongoing clinical trial with pazopanib. Three trials have been completed. Clinical trial PAZOGIST, phase II, multicenter study with 81 patients with metastatic and/or locally advanced unresectable GIST resistant to imatinib and sunitinib had promising results [71].

Another drug, crenolanib is a PDGFRA inhibitor. There is an ongoing clinical trial phase III with 120 patients with advanced or metastatic GIST, with a D842V mutation in the PDGFRA gene, without any results as of yet [72].

Ponatinib is a TKI inhibitor1. It is used in an ongoing clinical trial phase II, with 81 participants with metastatic or unresectable GIST, as a second-line therapy after treatment with imatinib, without any results as of yet [73].

Moreover, Dasatinib is an inhibitor of the Src-family protein-tyrosine kinases. It was used in one clinical trial with ipilimumab and first-line treatment without good results [74]. Apart from that, there is no clinical trial using this molecule.

Furthermore, binimetinib is a MEK1/2 inhibitor. An ongoing phase Ib/II clinical trial with 62 patients with advanced GIST combines it with imatinib, without results as of yet [75]. Another clinical trial, which will start recruiting patients soon, will combine binimetinib with pexidartinib [76].

Vandetanib is a VEGFR2 inhibitor. It is used in a phase II clinical trial in children and adults, with only nine participants with wild type GIST. It is a completed clinical trial and the results are about to be published [77].

The molecule famitinib is an receptor tyrosine kinase (RTK) inhibitor targeting c-Kit, VEGFR2, PDGFR, VEGFR3, Flt1, and Flt3. It is used in phase II clinical trial, 88 patients with advanced or metastatic GIST, as second-line therapy after imatinib, with no available results as of yet [78]. Another phase III clinical trial with advanced GIST patients, after imatinib’s failure, as a second-line therapy, compared to sunitinib [79].

Anlotinib is an RTK inhibitor VEGFR2 and VEGFR3 inhibitor. There is one ongoing phase III clinical trial with patients with advanced GIST after imatinib failure. No results are known as of yet [80].

Moreover, Axitinib is a PDGFR inhibitor. There is an ongoing single-arm, phase II (AXAGIST) clinical trial combining avelumab with axitinib in patients with unresectable or metastatic GIST, as a second-line treatment, with no known results as of yet [56]. Another recruiting observational clinical trial uses axitinib as second-line therapy for patients with metastatic renal cell carcinoma (mRCC), mantle cell Lymphoma (MCL), and GIST; no results are available [81].

Molecule alvocidib is a cyclin-dependent kinase (CDKs) inhibitor. A completed phase I clinical trial with 36 patients with metastatic or recurrent GISTs and sarcomas, combined with doxorubicin, with poor results, unfortunately [82].

Ribociclib is a CDK inhibitor. A recruiting Phase Ib, multicenter, open label study combines ribociclib with spartalizumab in patients with GIST, with no results yet [55].

Furthermore, Sorafenib is an RAF kinase inhibitor. A phase II clinical trial with imatinib and sunitinib treatment has failed, but there are no results yet [83]. Six clinical trials that have been completed in the past years have not shown encouraging results.

Moreover, pexidartinib is a TKI inhibitor. One clinical trial phase I is about to begin recruiting patients and will combine pexidartinib with binimetinib in patients with advanced GIST [76].

Olaratumab is a fully human IgG1 monoclonal antibody directed against PDGFRA, was given in one clinical trial, and the drug had an acceptable adverse events profile in patients with GIST. A phase II, open label, clinical trial, with 21 participants with previously treated unresectable or metastatic GIST. Although there was no apparent effect on PFS in patients without PDGFRA mutations, patients with PDGFRA-mutant GIST (all with D842V mutations) treated with olaratumab had longer disease control compared with historical data for this genotype [84].

Other molecules used: Masitinib TKR inhibitor was used in clinical trials with poor results; trametinib, a MEK MAPK/ERK kinase, was used in one clinical trial, which was withdrawn [85]. HQP1351 was used in a phase I recruiting clinical trial with patients with GIST [86], PLX9486 is a TKR inhibitor, and was used with or without sunitinib in patients with advanced solid tumors and GIST, a completed clinical trial without any results known [87]. Temozolomide is an alkylating agent used in central nervous system (CNS) cancers. It was used in a phase II clinical trial in patients with SDH-mutant/deficient GIST, with no available results [88]. DS-6157 is a monoclonal antibody that targets the G-protein coupled receptor 20 (GPR20). It was used in a phase I clinical trial in patients with GIST as a single therapy to participants [89]. In Figure 1, the link between molecules and signaling pathways in GISTs is represented.

## 5. Discussion

A very common problem with GIST therapy is resistance to imatinib. There have been a significant number of clinical trials trying to find new answers. Avapritinib is an effective solution, but only in PDGFRA D842V/exon 18 mutated GIST. Ripretinib challenges the therapeutic option in the second line and the results of the INTRIGUE trial are highly awaited. The other clinical trials performed are not able to give therapeutic solutions as of yet. There are 82 clinical trials, in all phases, active, and this is the first hopeful sign. The second hopeful sign is that the eight are using immunotherapeutic agents, such as nivolumab or ipilimumab, with or without chemotherapy. GISTs share some characteristics regarding the expression of PD1, PDL1, and immune checkpoint and membrane markers from immune cells, showing interesting data for the potential use of immunotherapy agents. Considering the impressive results immunotherapy has shown in other aspects of oncology, such as lung, renal, and melanoma, we are awaiting these studies’ results with great interest [90].

Resistance to imatinib, and in general TKI therapy, is a complex molecular process. Heterogeneity of secondary mutations is the principle mechanism of resistance and tumor progression. Several cellular signalings are involved, including Hippo pathway, MAPK, BET, PTEN, PI3K, PRKCQ, and JUN. Targeting these oncogenic mediators could potentially change the therapeutic approach of these untreatable tumors. Serrano et al., in their systematic approach, have suggested that a combination of TKIs might help improve the outcome of GIST tumors [25,26,91]. Further, Wozniak et al., in their review, recommend “liquid biopsy” as a potential mechanism of a personalized way to monitor resistance and response to TKI treatment [92].

Many drugs have been tested as treatment for GISTs. However, only four molecules (imatinib, dasatinib, axitinib, and ribociclib) have been combined with immunotherapy in clinical trials, showing there is a broad field for more studies, and that effort is being made to understand the molecular pathways of GISTs. The answer to the complex and persistent question of effective treatment of GISTs seems to include combination therapies. New small molecules, new TKIs, and immunotherapy agents, combined, may be the right way to inhibit the molecular pathways used by GIST cells to progress. Our review sheds light on the current research efforts, in both clinical and basic research, and this combined strategy could open new research pathways in disease therapy, and lead to the development of therapies that offer better clinical outcomes and expand life expectancy.

## Figures and Tables

**Figure 1 ijms-22-00493-f001:**
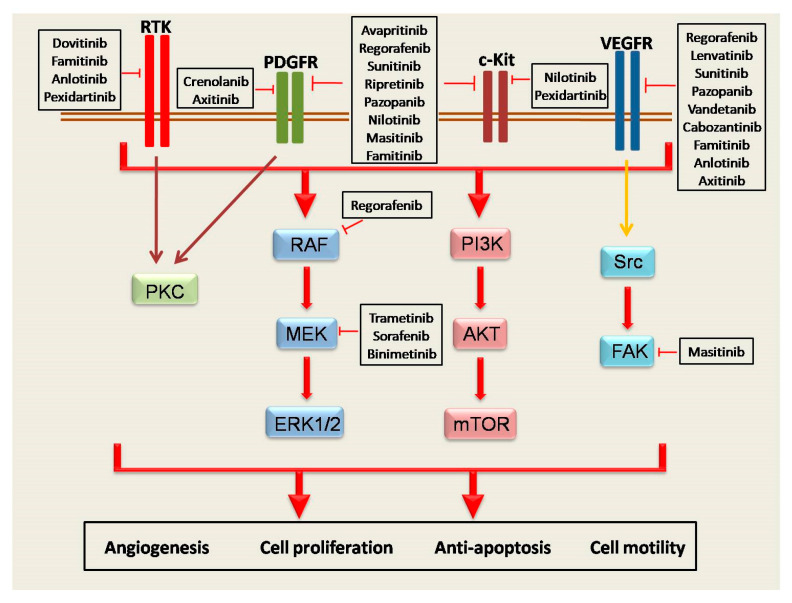
The role of the receptors in the activation of signaling pathways in GISTs. The four major receptors groups (RTKs, PDGFR, c-KIT, and VEGFR) closely regulate the cellular process such as angiogenesis, cell proliferation and motility, and anti-apoptotic capability of cancer cells through the induction of RAF/MEK/ERK, PI3K/AKT/mTOR, PKC, and Src/FAK axis. In addition, a plethora of small agents that directly target the receptors or signaling pathways is presented in the figure.

**Table 1 ijms-22-00493-t001:** Systemic Therapy for Resectable Gastrointestinal Stromal Tumors (GISTs) (NCCN Guidelines V.1.2021).

Neo Adjuvant Therapy	Adjuvant Therapy
Imatinib (sensitive mutations)	Imatinib
Avapritinib (PDGFR exon 18)	-

**Table 2 ijms-22-00493-t002:** Systemic Therapy for Unresectable GISTs (NCCN Guidelines V.1.2021).

First Line	Second Line	Third Line	Fourth Line	Additional Options
Imatinib	Sunitinib	Regorafenib	Ripretinib	Avapritinib
Avapritinib	-	-	-	Cabozantinib
-	-	-	-	Dasatinib
-	-	-	-	Everolimus
-	-	-	-	Nilotinib
	-	-	-	Pazopanib
-	-	-	-	Sorafenib
-	-	-	-	Larotrectinib

**Table 3 ijms-22-00493-t003:** Imatinib ongoing clinical trials.

Clinical Trial	Combination	Phase
NCT04006769	Entacapone	I
NCT02924714	none	N/A
NCT01991379	MEK 162	I/II
NCT02365441	Regorafenib	II
NCT03609424	PDR001	I/II
NCT01541709	none	II
NCT03862768	Sunitinib	N/A
NCT02260505	none	III
NCT04138381	Selinexor	I/II
NCT04193553	Lenvatinib	II
NCT03944304	Paclitaxel	II
NCT01742299	none	IV
NCT03381053	none	IV
NCT02413736	none	III
NCT01738139	Ipilimumab	I

**Table 4 ijms-22-00493-t004:** Sunitinib ongoing clinical trials.

Clinical Trial	Combination	Phase
NCT02164240	Regorafenib	I
NCT04633122	Ripretinib	II
NCT03673501	Ripretinib	III
NCT04409223	Famitinib	III
NCT01694277	Masitinib	III
NCT03862768	Imatinib	N/A
NCT00700258	Axitinib/Temsirolimus	III

**Table 5 ijms-22-00493-t005:** Regorafenib ongoing clinical trials.

Clinical Trial	Combination	Phase
NCT03465722	Avapritinib	III
NCT02164240	Sunitinib	I
NCT02638766	none	II
NCT02365441	imatinib	II
NCT01933958	none	N/A
NCT03475953	Avelumab	I/II
NCT03890731	none	II

**Table 6 ijms-22-00493-t006:** Immunotherapy-clinical trials.

Immunotherapeutic Agent	Clinical Trials	Active	Completed	Target
Ipilimumab	6	5	1	CTLA-4
Nivolumab	4	4	0	PD-1
Spartalizumab	1	1	0	PD-1
Pembrolizumab	3	2	1	PD-1
PDR001	1	1	0	PD-1
Avelumab	2	2	0	PD-1

**Table 7 ijms-22-00493-t007:** Immunotherapeutic Agents in ongoing clinical trials.

Molecule	Clinical Trial	Phase
Ipilimumab/Nivolumab	NCT02880020	II
Ipilimumab	NCT01738139	I
Ipilimumab/Nivolumab	NCT02500797	II
Ipilimumab/Nivolumab	NCT02834013	II
Ipilimumab/Nivolumab	NCT02982486	II
Spartalizumab	NCT04000529	I
PDR001	NCT03609424	I/II
Avelumab + Regorafenib	NCT03475953	I/II
Avelumab + Axitinib	NCT04258956	II
Pembrolizumab + Epacadostat	NCT03291054	II
	NCT02406781	II

## Data Availability

No new data were created or analyzed in this study. Data sharing is not applicable to this article.

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
