# Peer review of "Gastrointestinal Stromal Tumors (GISTs): Novel Therapeutic Strategies with Immunotherapy and Small Molecules"

_ijms, 2021, doi:10.3390/ijms22020493_

Round 1

Reviewer 1 Report

Dear Editor, thank you so much for inviting me to revise this manuscript about the topic of novel treatment in the therapeutic scenario of GIST.

Understanding the role of novel therapies in this setting is a mandatory need.

In this evolving landscape, the study by Vallilas and colleagues comprehensively evaluate the current landscape of medical treatment in these malignancies, especially focusing on immunotherapy and novel straetegies.

On the basis of the above, it addresses a current topic.

The manuscript is quite well written and organized. English could be improved.

Figures and tables are comprehensive and clear.

The introduction explains in a clear and coherent manner the background of this study.

We suggest the following modifications:

  • Introduction section: although the authors correctly included important papers in this setting, we believe a couple of studies should be cited within the introduction (doi: 10.1080/13543784.2021.1857363.; doi: 10.1177/1758835920936932), only for a matter of consistency. We think it might be useful to introduce the topic of this interesting paper.
  • The authors should report more details regarding D842V mutant GISTs. In fact, patients with advanced D842V-mutant GIST have always been seen as the “black sheep” of GISTs, with poor prognosis, similar to all metastatic GISTs in the pre-imatinib era, and more details should be added. The fate of this genomically defined population has recently changed with the advent of avapritinib, a potent and highly selective KIT and PDGFRA type I inhibitor, showing for the first time prominent and durable responses, never seen before in D842V mutant GIST patients. In addition, in recent years the biological landscape of D842V mutant GIST has been deeply investigated for a better understanding of what moves under this peculiar subset of GIST and some promising insights have emerged. 
  • The authors should better explain some points regarding immunotherapy in GIST and recent studies regarding the immunogenicity of PDGFRA mutant GIST
  • Discussion section: Very interesting and timely discussion. Of note, the authors should expand the Discussion section, including a more personal perspective to reflect on. For example, they could answer the following questions – in order to facilitate the understanding of this complex topic to readers: What are the knowledge gaps and how do researchers tackle them? How do you see this area unfolding in the next 5 years? We think it would be extremely interesting for the readers, especially considering the landscape of medical treatment in GIST, where novel treatment options are opening the doors of a new world.

However, we think the authors should be acknowledged for their work. In fact, they correctly addressed an important topic in GIST management.

We believe this article is suitable for publication in the journal although major revisions are needed. The main strengths of this paper are that it addresses an interesting and very timely question and provides a clear answer, with some limitations. 

We suggest a linguistic revision and the addition of some references for a matter of consistency. Moreover, the authors should better clarify some points regarding medical treatment in GIST patients.

Author Response

Dear Editor, thank you so much for inviting me to revise this manuscript about the topic of novel treatment in the therapeutic scenario of GIST.

Understanding the role of novel therapies in this setting is a mandatory need.

In this evolving landscape, the study by Vallilas and colleagues comprehensively evaluate the current landscape of medical treatment in these malignancies, especially focusing on immunotherapy and novel straetegies.

On the basis of the above, it addresses a current topic.

The manuscript is quite well written and organized. English could be improved.

Figures and tables are comprehensive and clear.

The introduction explains in a clear and coherent manner the background of this study.

AUTHOR RESPONSE: Initially, we would like to thank the reviewer for his very fruitful and constructive remarks that aim to improve our manuscript. We would also like to thank the reviewer for his kind words that recognize our effort. Below we provide our detailed answers to each comment one-by-one to ensure the clarity of our statements.

Introduction section: although the authors correctly included important papers in this setting, we believe a couple of studies should be cited within the introduction (doi: 10.1080/13543784.2021.1857363.; doi: 10.1177/1758835920936932), only for a matter of consistency. We think it might be useful to introduce the topic of this interesting paper.

AUTHOR RESPONSE: We thank the reviewer for the comment and the interesting and relative papers. We added the references (Ref 18 and 20).

The authors should report more details regarding D842V mutant GISTs. In fact, patients with advanced D842V-mutant GIST have always been seen as the “black sheep” of GISTs, with poor prognosis, similar to all metastatic GISTs in the pre-imatinib era, and more details should be added. The fate of this genomically defined population has recently changed with the advent of avapritinib, a potent and highly selective KIT and PDGFRA type I inhibitor, showing for the first time prominent and durable responses, never seen before in D842V mutant GIST patients. In addition, in recent years the biological landscape of D842V mutant GIST has been deeply investigated for a better understanding of what moves under this peculiar subset of GIST and some promising insights have emerged.

The authors should better explain some points regarding immunotherapy in GIST and recent studies regarding the immunogenicity of PDGFRA mutant GIST

AUTHOR RESPONSE: We thank the reviewer. We analyzed further the section regarding D842V mutation and immunogenicity. Also, we added the proper citation. The changes in the manuscript are presented below:                                   “The D842V mutation is the most common mutation associated with primary resistance to imatinib because it alters the kinase domain formation and, therefore, negatively affects imatinib association [40][41] . In fact, the PFS of patients treated with imatinib carrying the D842V mutation compared to non-D842Vs is statistically significant (2.8 vs. 28.5 months, respectively) [42]. In another study of a cohort of patients with PDGFRA D842V-mutated GIST, imatinib resulted in progressive disease for most patients (58.9%), with PFS up to 8 months [43]. Also, resistance occurs due to this mutation in the sunitinib [40][41].

Many studies indicate that D824V mutation displays more immune cells with increased cytolytic activity [40][41][44][45][46][47]. In particular, express higher interferon levels and several chemokines, such as CXCL14; demonstrates additional driver-derived neoepitope-HLA binding proteins; and has more PD-1 and PD-L1 expressing tumors [40][41][47]. Differences in tumor microenvironment composition were also highlighted between the D824 mutation and the other GIST molecular subtypes. More specifically, there is an overexpression of T-regs and CD8 + T-cells and a lower CD4 + T-cell rate. Further, the T-cell-inflamed signature score for GISTs was at a similar rate with tumor types responsive to checkpoint inhibition [44][45]. All the above data positively support the implementation of an immune-treatment approach in GIST and, in fact, in patients carrying the D824V mutation."

About the avapritinib, we mention its role in page 7.

 Discussion section: Very interesting and timely discussion. Of note, the authors should expand the Discussion section, including a more personal perspective to reflect on. For example, they could answer the following questions – in order to facilitate the understanding of this complex topic to readers: What are the knowledge gaps and how do researchers tackle them? How do you see this area unfolding in the next 5 years? We think it would be extremely interesting for the readers, especially considering the landscape of medical treatment in GIST, where novel treatment options are opening the doors of a new world.

AUTHOR RESPONSE: We thank the reviewer for his remarks. In the discussion section, we tried to highlight better the prospects of treatment for GIST. The changes in the manuscript are presented below:

“GISTs share some characteristics regarding the expression of PD1, PDL1 and immune checkpoint and membrane markers from immune cells, showing interesting data for the potential use of immunotherapy agents.” “Resistance to imatinib and in general TKI therapy is a complex molecular process. Heterogeneity of secondary mutations is the principle mechanism of resistance and tumor progression. Several cellular signalings are involved, including Hippo pathway, MAPK, BET, PTEN, PI3K, PRKCQ and JUN. Targeting these oncogenic mediators could potentially change the therapeutic approach of these untreatable tumors. Serrano et al., in their systematic approach have suggested that combination of TKIs might help to improve the outcome of GIST tumors [25][94][95]. Further, Wozniak et al., in their review recommend “liquid biopsy” as a potential mechanism of a personalized way to monitor resistance and response to TKI treatment  [96].” “The answer to the complex and persistent question of effective treatment of GISTs, seems to include combination therapies. New small molecules, new TKIs and immunotherapy agents combined appropriately may be the right way to inhibit the molecular pathways used by GIST cells to progress. Our review has shed light on the current research efforts both in clinical and basic research and this combined strategy could open new research pathways in front of disease therapy and lead to the development of therapies that would offer a better clinical outcome and expand life expectancy.”

Reviewer 2 Report

This manuscript is a good description of the state of the art on novel therapeutic strategies for gastrointestinal stromal tumors (GIST). 

Author Response

This manuscript is a good description of the state of the art on novel therapeutic strategies for gastrointestinal stromal tumors (GIST).

AUTHOR RESPONSE: We would like to thank the reviewer for his kind words that recognize our effort.

Round 2

Reviewer 1 Report

The authors extensively modified the paper according to our suggestions. 

In particular, the authors provided more information in several sections, especially regarding immunotherapy and D842V.

One little flaw: we believe there has been a problem with bullet points, since every sections is numbered as 1.

In summary, the authors provided a comprehensive overview on several important clinical and research questions in this topic, and we believe they should be acknowledged for their work.

We recommend Acceptance in current form - after rechecking the number of the bullet points.